# The Influence of Diet and Physical Activity on Oxidative Stress in Romanian Females with Osteoarthritis

**DOI:** 10.3390/nu14194159

**Published:** 2022-10-07

**Authors:** Bogdana Adriana Nasui, Patricia Talaba, Gabriel Adrian Nasui, Dana Manuela Sirbu, Ileana Monica Borda, Anca Lucia Pop, Viorela Mihaela Ciortea, Laszlo Irsay, Anca Ileana Purcar-Popescu, Delia Cinteza, Madalina Gabriela Iliescu, Florina Ligia Popa, Soimita Mihaela Suciu, Rodica Ana Ungur

**Affiliations:** 1Department of Community Health, Faculty of Medicine, “Iuliu Hatieganu” University of Medicine and Pharmacy, Pasteur Street, No.4, 400349 Cluj-Napoca, Romania; 2Faculty of Law, “Dimitrie Cantemir” University, 60 Teodor Mihali Street, 400591 Cluj-Napoca, Romania; 3Department of Medical Specialties, Faculty of Medicine, “Iuliu-Hațieganu” University of Medicine and Pharmacy, 8 Victor Babeș Street, 400012 Cluj-Napoca, Romania; 4Department of Clinical Laboratory, Food Safety, “Carol Davila” University of Medicine and Pharmacy, 6 Traian Vuia Street, 020945 Bucharest, Romania; 5Department of Rehabilitation, Rehabilitation Clinical Hospital, 46-60 Viilor Street, 400066 Cluj-Napoca, Romania; 69th Department—Physical Medicine and Rehabilitation, “Carol Davila” University of Medicine and Pharmacy, 37 Dionisie Lupu Street, 020021 Bucharest, Romania; 7Department of Rehabilitation, Faculty of Medicine, Ovidius University of Constanta, 1 University Alley, Campus—Corp B, 900470 Constanta, Romania; 8Physical Medicine and Rehabilitation Department, Faculty of Medicine, “Lucian Blaga” University of Sibiu, Victoriei Blvd., 550024 Sibiu, Romania; 9Academic Emergency Hospital of Sibiu, Coposu Blvd., 550245 Sibiu, Romania; 10Department of Physiology, Faculty of Medicine, “Iuliu Hatieganu” University of Medicine and Pharmacy, 8 Victor Babes Street, 400012 Cluj-Napoca, Romania

**Keywords:** Romanian females, osteoarthritis, oxidative status, body mass index, diet, physical activity

## Abstract

Osteoarthritis (OA) is the most prevalent chronic joint disease, increases in prevalence with age, and affects most individuals over 65. The present study aimed to assess the oxidative status in relation to diet and physical activity in patients with OA. We used a cross-sectional study applied to 98 females with OA. Blood samples were collected to determine oxidative stress markers: malonyl dialdehyde (MDA), reduced glutathione (GSH), oxidized glutathione (GSSG), and GSH/GSSG. Diet was estimated with a standardized food frequency questionnaire. We used the International Physical Activity Questionnaire (IPAQ) to assess the females’ physical activity. Multiple regression analyses were executed to determine the association between the oxidative markers and the intake of vegetables and fruit. The study showed that most patients were overweight or obese (88.8%). The level of physical activity was above the recommended level for adults, mainly based on household activities. The intake of vegetables and fruit was low. The MDA marker was inversely, statistically significantly associated with the consumption of vegetables (*p* < 0.05). Public health policies must address modifiable risk factors to reduce energy intake and obesity and increase the intake of vegetables and fruit. Higher consumption of vegetables and fruit may provide natural antioxidants that can balance oxidative compounds.

## 1. Introduction

Osteoarthritis (OA) is the most prevalent chronic joint disease, increases in prevalence with age [1], and affects most individuals over 65 [2,3]; it is a disease of the entire joint, involving the cartilage, joint lining, ligaments, and bone. OA is characterized by the breakdown of the cartilage, bony changes of the joints, deterioration of tendons and ligaments, and various degrees of inflammation of the joint lining [4].

OA tends to affect commonly used joints, such as the hands and spine, and the weight-bearing joints, such as the hips and knees. OA symptoms include joint pain and stiffness [5,6].

OA multi-factorial etiology includes oxidative stress and the overproduction of reactive oxygen species (ROS), which regulate intracellular-signaling processes, chondrocyte senescence and apoptosis, extracellular matrix synthesis and degradation, and synovial inflammation and dysfunction of the subchondral bone [7].

The term oxidative stress represents the inbalance between the cells’ ROS and the cells’ antioxidant capacity. High levels of oxidative stress may damage the cells by oxidizing lipids and altering the DNA and protein structure [8]. 

ROS are free radicals containing oxygen molecules, including hydroxyl radical (OH^−^), hydrogen peroxide (H_2_O_2_), superoxide anion (O_2_^−^), nitric oxide (NO), and hypochlorite ion (OCl^−^). The presence of unpaired electrons in the valence shell causes ROS to be short-lived, unstable, and highly reactive to achieve stability [4]. The major sites of ROS generation include the mitochondria (through oxidative phosphorylation), non-mitochondrial membrane-bound nicotinamide adenine dinucleotide phosphate (NADPH) oxidase, and xanthine oxidase (XO). The body has multiple mechanisms for scavenging ROS; the antioxidant system includes enzymatic and non-enzymatic antioxidants, such as superoxide dismutase (SOD), catalase (CAT), glutathione peroxidase (GPX), glutathione (GSH), NADPH ubiquinone oxidoreductase (NQO1), paraoxonases (PON), and natural antioxidants such as ascorbic acid (vitamin C), α-tocopherol (vitamin E), and carotenoids [9,10].

Obesity is linked with OA through (1) mechanical stress and (2) the adipokine system. Recent epidemiological data showed an increased risk of hand osteoarthritis in obese patients, underlying the role of the systemic inflammatory system [11] and adipokines, released by adipose tissue [12,13].

Environmental factors influencing the prevalence of OA are modifiable risk factors and include occupation, diet, level of physical activity, and obesity [14,15]. 

In our study, we chose to determine the following oxidative status markers: malonyl-dialdehyde (MDA); glutathione, and the ratio of GSH/GSSG. MDA is the primary marker of lipid peroxidation. The prime targets of ROS attack are the polyunsaturated fatty acids in the membrane lipids causing lipid peroxidation (LPO), which may lead to disorganization of the cell structure and function. Further decomposition of peroxidized lipids yields various end-products, including malondialdehyde (MDA) [16]. The measurement of MDA is widely used as an indicator of LPO [17].

Glutathione is an antioxidant produced in cells, with a structure of tripeptide (cysteine, glycine, and glutamic acid) existing either in a reduced (GSH) or oxidized (GSSG) form [18]. It is produced by the liver and is involved in many processes. Experimental studies on rats showed that mechanical stress promotes the accumulation of reactive oxygen species (ROS) in chondrocytes in vivo, resulting in chondrocyte apoptosis and leading to osteoarthritis development in a rat model. Osteoarthritis development was inhibited by oral administration of N-acetyl cysteine (NAC), and it was demonstrated to have efficacy in reducing cartilage degradation and inflammation markers as well as significant improvements in pain and functional outcomes [19,20].

Few studies in the literature on human subjects have investigated oxidative markers from blood samples; to our knowledge, most of the studies focused on preclinical models. 

Less is known about the relationship between the oxidative status markers, diet and lifestyle in osteoarthritis patients. Our study aimed to assess the oxidative status (lipid peroxidation and antioxidant status) in relation to the diet and physical activity of patients with OA. To our knowledge, this is the first study in Romania that established a relationship between the oxidative status of the body reflected in the biochemical parameters, physical activity, and diet. Because diet and lifestyle are related to body mass index, the second objective of our study was to establish a relationship between weight status and oxidative markers in patients with OA.

## 2. Materials and Methods

We used a cross-sectional study on 98 female osteoarthritis patients diagnosed according to the American College of Rheumatology (ACR) criteria for the classification of OA. The data were collected from July 2019 to December 2019. The exclusion criteria were: refusal of patients to participate in the study; patients with psychiatric disorders; VSH (sedimentation rate) > 30 mm/h, CRP > 1.5 (C-Reactive Protein); inflammatory arthritis; and crystal arthritis. The diagram flow of the sample selection is presented in Figure 1.

The 5 mL peripheric blood samples were collected from patients after an overnight fast (10–12 h) in two ethylenediaminetetraacetic acid tubes. MDA is a stable product of the reaction between oxidative radicals and lipids [3]. Free plasma MDA was determined using the spectrofluorimetric method described by Conti [21]. This method is based on the reaction between MDA and thiobarbituric acid with fluorescent adduct synthesis proportionally to MDA concentration. A Perkin Elmer spectrofluorometer was used for emission intensity measurement at 534 nm in the synchronous fluorescence system at a 14 nm wavelength difference (Δλ) between excitation and emission. A calibration curve performed with known MDA concentrations was used. The concentration values were expressed in nmol/mL.

### 2.1. The GSH/GSSG Ratio Assessment

The GSH/GSSG ratio was calculated using the method proposed by Vats [22]. A total of 50 mL of blood was treated with 450 mL of 10% solution of m-phosphoric acid and centrifuged for 10 min at 1000× *g*. For the determination of GSH, 0.1 mL of the supernatant was diluted with 1.8 mL of 0.1 M phosphate buffer (pH 8) continuing 5 nmol/L EDTA after which 0.1 mL solution of o-phthalaldehyde in methanol (1 mg/mL) was added. After 15 min of incubation, the fluorescence emitted at 420 nm was measured with an excitation of 350 nm. The GSSG was estimated in 250 μL supernatant which was incubated for 30 min with 40 nmol/L N-ethylmaleimide. Then, 0.65 mL 0.1N NaOH was added, followed by adding 0.1 mL reaction mixture formed by 1.8 mL 0.1N NaOH and 0.1 mL o-phthalaldehyde solution. 

The GSH and GSSG concentration calculation was based on calibration curves obtained with GSH and GSSG known concentrations and processed similarly. The GSH ratio was calculated.

### 2.2. Questionnaire and Variables

The research team applied a structured questionnaire to the study’s participants consisting of (1) a demographic section with a personal medical history and (2) a validated food frequency questionnaire. We used a food frequency questionnaire section to assess the dietary intake of vegetables and fruit. The food frequency questionnaire (FFQ) is part of a valid questionnaire adapted to Romanian habits (EPIC Norfolk). We tested and re-tested the questionnaire on a sample of 30 respondents and calculated Spearman’s correlation coefficient to assess the reliability (r = 0.766). The questionnaires were administered by interview. We calculated Kappa statistics to assess the inter-rater agreement; the kappa result was 0.8.

In the FFQ section, we assessed the food and beverages intake-habits as different frequencies: never or less than 1/month; 1–3 times/month; once/week; 2–4 times/week; 5–6 times/week; once/day; 2–3 times/day; 4–5 times/day; more than six times/day. A portion of fruit/vegetables was considered to be 80 g of a medium-size fruit, 1/2 cup of chopped/cooked vegetables, 30 g of dried fruit, or 150 mL of fruit or vegetable juice.

Anthropometric parameters weight and height were measured during the hospitalization. Body mass index (BMI) was calculated “as weight (kg) divided by the square of height (m^2^)”, and accordingly, obesity was defined from a BMI of >30 kg/m^2^. The nutritious status of the patients was divided as underweight BMI < 18.5 kg/m^2^; normoweight: BMI = 18.5–24.9 kg/m^2^; overweight: BMI = 25–29.9 kg/m^2^; obesity type 1 BMI = 30–34.9 kg/m^2^; obesity type 2 BMI = 35–39.9 kg/m^2^; and type 3 BMI > 40 kg/m^2^.

Physical activity was assessed using the International Physical Activity Questionnaire (IPAQ) [23] as duration, frequency, and intensity. Vigorous physical activities involve hard physical effort and render breathing much harder than usual. Moderate activities require moderate physical effort and render breathing harder than usual. We estimated the job-related physical activity, transportation, household physical activity, and leisure time. We calculated the total physical activity in minutes per day.

### 2.3. Statistical Analyses

The data were analyzed using the Statistical Package for the Social Sciences (SPSS), version 20. Descriptive statistics were calculated: means and standard deviations (SD) for continuous variables or frequencies and percentages for categorical variables. The sample size was calculated according to the Cochran formula, taking into account the prevalence of OA in the Romanian population (6.43% prevalence, the result was 92.45).

Multiple regression was run to predict the MDA, GSSH, and GSSG association with dietary vegetables and fruit intake.

## 3. Results

All of the participants in the study were selected to be females (*n* = 98, 100%) from Romania. The studied sample’s mean age was 65.57 ± 0.67 years. The mean age for menopause was 47.13 ± 0.57 years, and 75.6% (*n* = 68) were under 50; most of the patients lived in rural areas, 52.1% (*n* = 49).

Most of them had a secondary level of education, 47.3% (*n* = 44). Most had a hereditary history of osteoarthritis 63.3% (*n =* 63) (Table 1).

We calculated the body mass index (BMI) depending on the classes of obesity. According to the results of our study, only 11.2 % (*n* = 10) of females were normal weight; the rest of them were overweight (34.69%, *n* = 34), or obese (45.91%, *n* = 45) (Table 2).

### 3.1. The Oxidative Stress Markers of the Sample

The mean values of oxidative markers of Romanian female patients with osteoarthritis are presented in Figure 2 below. The ratio GSH/GSSG was 9.19 ± 5.11.

The study analyzed the oxidative stress markers depending on the body mass index of Romanian female patients with OA. The results showed no statistical differences between females with different nutritional statuses (Table 3).

### 3.2. Food Behavior among the Study Group

Regarding food behavior, we analyzed the intake of raw and boiled vegetables and fruit. The present study showed the daily consumption of a suboptimal amount of vegetables: only 21.4% (*n* = 21) of patients ate a portion of raw vegetables daily, and only 16.3% (*n* = 16) of females with OA ate a portion of boiled vegetables daily. However, the daily fruit consumption was higher, 60.2% (*n* = 59) of the females ate at least one portion of fruit daily (Table 4).

We analyzed the level of oxidative stress markers of the body (MDA, GSH, GSSG, and GSH/GSSG) depending on raw vegetable consumption. The results of the present study showed no statistical differences depending on the frequency of consumption (Table 5).

The study estimated the plasma level of lipid peroxidation, MDA, and antioxidant compounds’ activities, GSH, GSSG, and GSH/GSSG, depending on the frequency of boiled vegetable intake. Our results showed no statistically significant differences in the plasma level of oxidative status markers depending on the amount of boiled vegetables consumed (Table 6).

The study assessed the frequency of fruit intake and the plasma levels of oxidative stress markers in the body. The study showed no statistically significant increase in antioxidant markers with the increased consumption of fruit (Table 7).

### 3.3. Physical Activity of the Sample

The present study analyzed the level of physical activity of the patients with OA. As the study results showed, all of the female patients met the recommended level of physical activity per day. The physical activity was mainly comprised of at-home moderate physical activity, such as gardening and rural farm work (Table 8), and much lesser professional (organized) physical activity (Table 9). There are no statistical differences regarding the level of physical activity depending on nutritional status.

Professional physical activity was performed at a lower amount than home physical activity since most of the female patients with OA were retired (the mean age of the sample was 65 years) (Table 9).

The total physical activity of the sample exceeds the recommended level for a healthy lifestyle. There were no statistical differences between the different nutritional statuses of the patients and the amount of physical activity (*p* = 0.272) (Figure 3).

### 3.4. Correlates between Oxidative Markers and Vegetables and Fruit Intake

To predict the association of the enzymatic status of the body with the antioxidants nutrients from food intake from vegetables and fruit, we run a multivariate logistic regression. In this model, we considered the frequency 0 (from never to 1/day intake (exclusive)) and frequency one from 1/day to 2–3 times/week). The study showed that the oxidative marker MDA is statistically significantly inversely associated with vegetables and fruit intake (Table 10).

The multiple regression model to predict the association between GSH and vegetables and fruit consumption showed a direct correlation, but the results were not statistically significant (Table 11).

The same model was run to predict the level of GSSG and the intake of vegetables and fruit, but the results showed no statistical differences (Table 12).

The multiple regression analysis was performed to predict the plasma levels markers of oxidative status and the ratio of GSH/GSSG depending on the intake of vegetables and fruit. No statistically different associations were found between fruit and vegetable consumption and the GSH/GSSG ratio (Table 13).

## 4. Discussion

Our study aimed to assess the oxidative status (lipid peroxidation and antioxidant status) in relation to the diet and physical activity of patients with OA.

Obese individuals have shown markers indicative of oxidative stress: (1) elevated levels of reactive oxygen species and (2) diminished antioxidant defenses, which are associated with lower antioxidant enzymes [24]. Our study showed that 89.9% of the investigated patients with OA were overweight or obese. The pathophysiology of the link between OA and obesity is related to the direct effect of excess mechanical loads being placed on the cartilage and an adipose tissue effect [25]. Adipocytes produce and release adipokines (e.g., leptin). They are also a local inflammatory reaction site when the adipose tissue is ectopic. Each additional kilogram of body weight adds a six-kilogram load to each knee [26,27]. This excess weight can induce cartilage degeneration because of more significant mechanical stress on weight-bearing joints. Leptin is a cytokine produced by adipocytes in white adipose tissue (hence its name “adipokine”) [28], which is released into the systemic circulation where it can reach the joints through the subchondral vascular network [29,30]. Chondrocytes have leptin receptors; the adipokines play an essential role in cartilage and bone homeostasis, but at overly high concentrations, they contribute to the appearance and progression of OA (destruction of cartilage) [31]. A higher leptin concentration has been found in the synovial fluid of arthritic joints than that of non-arthritic joints [32]. Visceral adipose tissue is significantly reduced with regular physical activity (following the health guidelines for 150 min of moderate physical activity per week), even if no weight is lost [33]. Despite the existing literature data regarding the link between obesity, OA, and oxidative status, the present study cannot support these due to the small sample size, with a small percentage of normoweight women. Future studies, of larger sample sizes of Romanian patients with OA, are required to establish this relationship.

Low education, non-managerial occupation, and income level tend to predict pain, physical dysfunction, and disability among adults with osteoarthritis. These associations can be attributed to more significant strenuous physical activity among people with low social and economic status [34]. Low education attainment and occupation are associated with low socioeconomic status (SES) and the development of other chronic illnesses. In the present study, 84.9% of the patients with osteoarthritis had primary or secondary education. Moreover, 52.1% were living in rural areas. The physical activity assessment of females with OA showed an active life with a level over the minimum recommendation for healthy adults. This level is attributed mainly to moderate home physical activity, evidenced by our study. Females with low education and the majority from the rural area may be involved in more physical activity (mainly at home, non-organized) than other females. People from rural areas are also more likely to be involved in physical activities such as gardening. There are no differences in the level of physical activity between the categories of obesity among patients with OA; PA could lead to a protective effect. The results also showed no differences in the levels of enzymes in patients with different degrees of obesity. Low socioeconomic status may influence their behaviors regarding seeking and access to medical care. In addition, these conditions may impact on health outcomes, quality of life, and other chronic conditions and mortality [35]. 

Both aerobic and anaerobic activities possess the potential to result in increased ROS and RNS production and subsequent oxidative stress. While obesity has been shown to exacerbate the oxidative stress response, dietary manipulation and exercise training may serve as an effective intervention to ameliorate oxidative stress profiles. Whether exercise training improves oxidative stress and inflammatory profiles in the absence of weight loss remains unclear; however, strict calorie restriction alone or coupled with physical activity intervention demonstrates promise in alleviating oxidative stress in obese individuals when accompanied by weight reduction [36,37]. On the other hand, animal models showed that exercise modulates transcription in several metabolic pathways associated with extracellular matrix (ECM) biosynthesis and inflammation/immune responses in the normal cartilage of rats undergoing treadmill walking. Moreover, exercise was able to suppress the expression of genes involved in ECM degradation, bone formation, and initiation of pro-inflammatory cascades, which are known to be upregulated in OA (Mmp9, Mmp8, Igf1, ColIa1, Adamts3, Adamts14), highlighting a positive effect on cartilage preservation [38].

Since our sample comprised mainly patients with obesity or overweight, we cannot draw clear conclusions regarding the relationship between the level of PA, obesity and oxidative status, despite other studies that showed that exercise improved the MDA oxidative stress marker [39]. Exercise training improves the antioxidant enzyme activity with no telomere length changes [40]. The results of the present study showed that the oxidative stress (oxidant and antioxidant balance) was not statistically different in patients with obesity and patients with different levels of physical activity.

Fruit and vegetables are sources of natural antioxidants such as vitamin C, vitamin E, carotenoids, and flavonoids. However, the protective effects against diseases may also be the result of unknown antioxidant compounds or the synergy of several different antioxidants in fruit and vegetables [41,42]. Fruit and vegetables are a source of glutathione and increase the erythrocytes’ glutathione peroxidase activity and resistance of plasma lipoproteins to oxidation [43].

The results of our study showed a low intake of vegetables in the study group. The results are similar to other studies on different samples of the Romanian population, which showed that the vegetable intake is low compared with the recommendation [44,45]. On the other hand, fruit are consumed in higher amounts than vegetables.

Patients with OA may have a lower level of plasma antioxidant status due to the reduced intake of vegetables and fruit. The multivariate analyses showed no statistically significant relationship between vegetable and fruit intake and the body’s antioxidant status (GSH, GSSG, GSH/GSSG). A high-calorie diet also correlates with a need for a diet enriched with antioxidants. 

Our study showed that only 11% of our patients were normal weight; the rest were overweight or obese. These results suggest that OA patients may have a high energy intake, taking into account the level of physical activity of the sample.

Increased consumption of vegetables and fruit would reduce oxidative stress and damage markers, but it is unknown whether these markers are related to dietary intake [46,47,48]. 

ROS are known to be involved in biological events such as cell signaling, chondrocyte senescence, apoptosis, extracellular matrix synthesis and degradation, synovial inflammation, and dysfunction of subchondral bone. ROS have been recognized as signaling molecules that regulate a variety of physiological processes, for example, are required for cytokines, insulin, growth factor, and AP-1 and NF-kB signaling. The antioxidant property of GSH is mostly attributed to GSH peroxidase-catalyzed reactions, where hydrogen peroxide and lipid peroxide are reduced while GSH is oxidized to GSSG. GSSG is then reduced back to GSH by GSSG reductase via the utilization of nicotinamide adenin dinucleotide phosphate (NADPH) establishing the redox cycle. GSH largely determines the intracellular redox potential: the GSSG ratio and oxidative stress overcome the ability of the cell to reduce GSSG in GSH [20].

A recent discovery showed that OA pathogenesis was linked with a form of regulated cell death named ferroptosis. Ferroptosis is characterized by the iron-dependent accumulation of lipid hydroperoxide that reaches lethal levels. Yao et al. [49] first indicated that chondrocytes underwent ferroptosis under inflammatory and iron-overload conditions and that ferroptosis contributed to the progression of OA in vivo and promoted matrix metalloproteinase (MMP)-13 expression while inhibiting COL2 expression in chondrocytes cultured in vitro. Miao et al. [50] found that iron accumulated in cartilage and synovial fluid during OA progression and that the expression of biomarkers of the peroxidation defense system, including glutathione peroxidase (GPX4) and glutathione (GSH) levels, was decreased in the patient samples. Moreover, as a characteristic change in ferroptosis, morphological changes in mitochondria have also been observed in OA cartilage by transmission electron microscopy, indicating that ferroptosis is closely associated with OA [51].

The present study showed that MDA was significantly inversely associated with a low intake of vegetables and fruit consumption. Similar to our findings, studies from the literature showed that vegetables and fruit consumption enhanced antioxidant levels (SOD and GPX) and reduced oxidant levels (MDA) and genetic damage [52,53]. 

There are few studies in the literature on human data that investigated oxidative markers from blood samples [10]. 

To the best of our knowledge, this is the first study in Romania to analyze the level of GSH and GSSG from blood samples of patients with OA. Unfortunately, our results cannot be compared to data on human models because the majority of the studies are preclinical. The ratio GSH:GSSG is a better marker of oxidative stress compared to antioxidant enzymes from blood samples that can be consumed in redox reaction [54]. 

Our study showed the need for the embedding of public health policies in order to improve the lifestyle of patients with OA. Females must be educated to increase vegetable and fruit consumption. In addition, one must be educated to reduce caloric intake to prevent overweight and obesity. By having healthy food choices with an increased vegetable intake, patients with OA may reduce the caloric-dense food consumption. On the other hand, maintaining an active lifestyle will increase energy expenditure, reduce adipose tissue, and decrease adipokine. In addition, supplementation with GSH exogenously (as a precursor—N-acetyl-cysteine) by functional foods or supplements could be an opportunity to improve the antioxidant status in OA; however, further interventional studies are needed on this topic.

Limitations. Our study is subject to limitations. First is the size of the sample. Larger sample sizes would provide better information. However, according to statistical calculation, the sample represented the population. Another limitation of the study was the reported questionnaire—both diet and physical activity data were collected by interview. Bias can appear by overestimation or underestimation of different categories of food intake in the FFQ. Physical activity may also be overestimated; future objective methods may be used to estimate physical activity or diet better.

## 5. Conclusions

The present study’s results showed that most female patients with OA were overweight or obese. The respondents’ physical activity met the recommended level, mainly due to homework activities. Females with OA had a reduced intake of vegetables and fruit. MDA, the marker of oxidative stress, was statistically inversely correlated with the intake of vegetables and fruit. Thus, public health policies must be implemented targeting modifiable risk factors—reducing energy intake and obesity and increasing the intake of vegetables and fruit. A higher consumption of vegetables and fruit may provide natural antioxidants that can balance oxidative compounds. 

## Figures and Tables

**Figure 1 nutrients-14-04159-f001:**
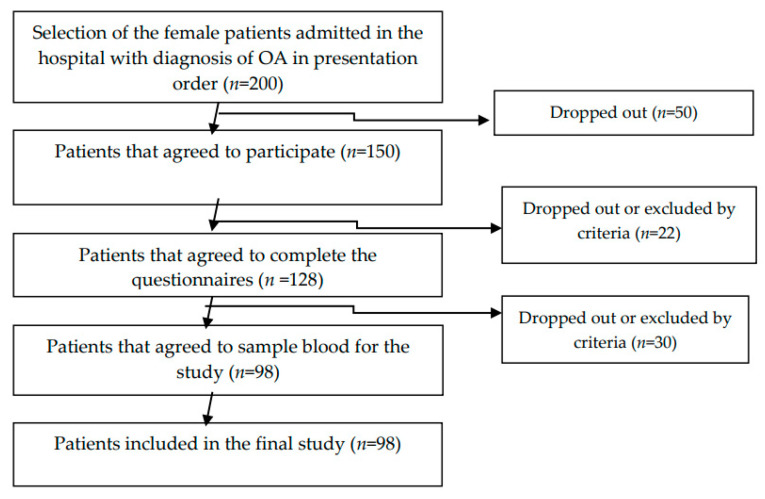
Reporting flow diagram (guideline).

**Figure 2 nutrients-14-04159-f002:**
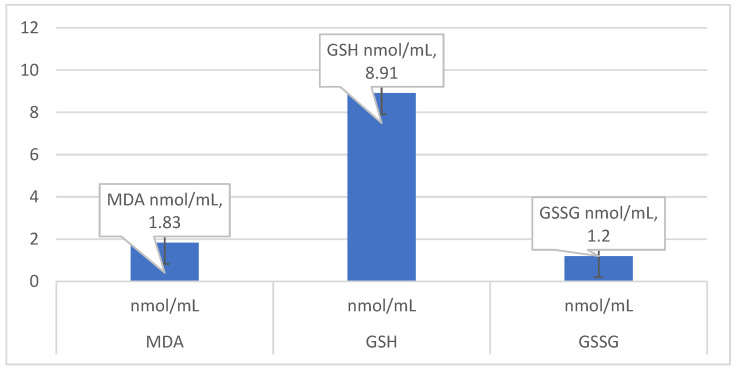
Mean values of the status of the oxidative marker in the sample.

**Figure 3 nutrients-14-04159-f003:**
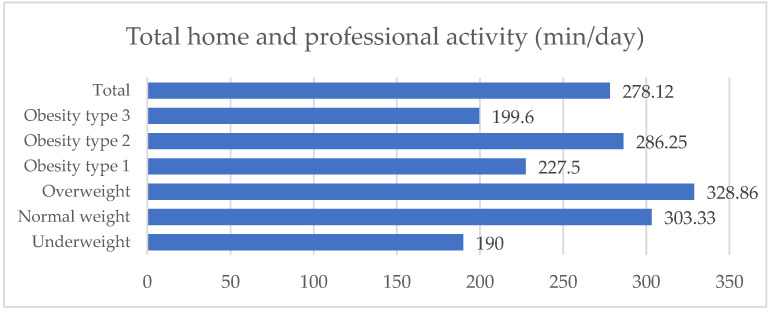
The total physical activity of the sample.

**Table 1 nutrients-14-04159-t001:** Demographic data of the sample.

Variable		
Mean Age		65.57 ± 0.67
Mean Menopause age		47.13 ± 0.57
Menopause	Yes	94 (95.9%)
	No	4 (4.1%)
Menopausal age	Under 50 years	68 (75.6%)
	Over 50 years	22 (24.4%)
Residence	Urban	45 (47.9%)
	Rural	49 (52.1%)
Education	Primary	35 (37.6%)
	Secondary	44 (47.3%)
	Higher	14 (15.1%)
Family history of OA	Yes	33 (35.5%)
	No	60 (64.5%)
Smoker status	Yes	1 (1.1%)
	No	86 (91.5%)

**Table 2 nutrients-14-04159-t002:** Nutritional status of the sample depending on BMI.

Variable BMI kg/m^2^	Number (%)
Underweight	9 (9.18%)
Normal weight	10 (10.20%)
Overweight	34 (34.69%)
Obese Type 1	24 (24.49%)
Obese Type 2	11 (11.22%)
Obese Type 3	10 (10.20%)

**Table 3 nutrients-14-04159-t003:** The level of oxidant status markers of the sample depending on the BMI.

Variable	MDA nmol/mL	GSH nmol/mL	GSSG nmol/mL	GSH/GSSG
Underweight	1.57 ± 0.55	9.26 ± 0.70	1.04 ± 0.45	9.92 ± 5.013
Normal weight	1.68 ± 0.49	9.10 ± 1.94	1.13 ± 0.414	9.52 ± 5.10
Overweight	1.83 ± 0.69	9.38 ± 1.60	1.17 ± 0.48	10.10 ± 5.93
Obese Type 1	1.75 ± 0.45	8.45 ± 1.13	1.24 ± 0.44	8.13 ± 4.10
Obese Type 2	1.96 ± 0.71	8.89 ± 1.48	1.21 ± 0.45	8.95 ± 5.05
Obese Type 3	2.09 ± 0.63	8.40 ± 1.72	1.38 ± 0.49	7.49 ± 4.65
*p* value	0.58	0.25	0.82	0.66

Data are expressed as mean ± SD, ANOVA, *p* < 0.05 is statistically significant.

**Table 4 nutrients-14-04159-t004:** Frequency of vegetable and fruit frequency intake (one portion).

Variable	Never	Monthly	Once a Week	2–4 Times a Week	5–6 Times a Week	Daily
Raw vegetables	5 (5.1%)	6 (6.1%)	18 (18.4%)	37 (37.8%)	11 (11.2%)	21 (21.4%)
Boiled vegetables	7 (7.1%)	4 (4.1%)	13 (13.3.)	35 (35.7%)	23 (23.5%)	16 (16.3%)
Fruit	0	0	4 (4.1%)	21 (21.4%)	14 (14.3%)	59 (60.2%)

**Table 5 nutrients-14-04159-t005:** The oxidative stress markers and the frequency of raw vegetable intake.

Variable		Raw Vegetables						
		Never	Monthly	Once/Week	2–4 Times/Week	5–6 Times/Week	Daily	*p*-Value
MDA	(nmol/mL)	1.61 ± 0.28	1.87 ± 0.57	1.82 ± 0.55	1.92 ± 0.69	2.05 ± 0.62	1.57 ± 0.48	0.226
GSH		9.34 ± 2.28	8.33 ± 0.83	9 ± 1.38	8.68 ± 1.58	8.64 ± 1.08	9.43 ± 1.79	0.453
GSSG		1.12 ± 0.42	1.29 ± 0.49	1.28 ± 0.43	1.20 ± 0.50	1.36 ± 0.42	1.36 ± 0.42	0.551
GSH/GSSG		9.49 ± 4.51	7.90 ± 4.84	8.26 ± 4.56	9.28 ± 5.60	7.65 ± 5.08	7.65 ± 5.08	0.662

Data are expressed as mean ± SD, ANOVA, *p* < 0.05 is statistically significant.

**Table 6 nutrients-14-04159-t006:** The oxidative stress markers and the frequency of boiled vegetables intake.

Variable		Never	Monthly	Once a Week	2–4 Times/Week	5–6 Times/Week	Daily	*p*-Value
MDA	nmol/mL	1.60 ± 0.70	1.96 ± 0.89	1.72 ± 0.55	1.81 ± 0.60	1.95 ± 0.61	1.80 ± 0.56	0.785
GSH		9.24 ± 1.35	8.23 ± 1.44	9.20 ± 1.59	8.97 ± 1.36	8.42 ± 1.19	9.27 ± 2.32	0.457
GSSG		1.04 ± 0.46	1.50 ± 0.38	1.14 ± 0.40	1.17 ± 0.48	1.17 ± 0.48	1.17 ± 0.48	0.609
GSH/GSSG		10.60 ± 5.18	6.03 ± 2.98	9.11 ± 3.71	9.77 ± 5.84	8.05 ± 4.36	9.21 ± 5.77	0.611

Data are expressed as mean ± SD, ANOVA, *p* < 0.05 is statistically significant.

**Table 7 nutrients-14-04159-t007:** The oxidative stress markers and the frequency of fruit intake.

Variable		Once a Week	2–4 Times/Week	5–6 Times/ Week	Daily	*p*-Value
MDA	nmol/mL	1.80 ± 0.92	2.03 ± 0.65	1.93 ± 0.65	1.72 ± 0.54	0.196
GSH		7.79 ± 1.45	9.00 ± 1.73	8.43 ± 1.32	9.06 ± 1.52	0.260
GSSG		1.26 ± 0.68	1.28 ± 0.48	1.28 ± 0.51	1.17 ± 0.43	0.712
GSH/GSSG		8.28 ± 5.32	9.03 ± 6.04	8.64 ± 6.05	9.28 ± 4.60	0.961

Data are expressed as mean ± SD, ANOVA, *p* < 0.05 is statistically significant.

**Table 8 nutrients-14-04159-t008:** Physical activity of sample—at-home routine work activity.

Variable PA (min/day ± DS)	Body Mass Index Category (kg/m^2^)							
	Underweight	Normal Weight	Overweight	Obesity Type 1	Obesity Type 2	Obesity Type 3	Total	*p* Value
Home intense PA	30 ± 42.42	32.50 ± 53.44	60 ± 116.41	6.25 ± 17.64	1.25 ± 4.33	30 ± 94.86	31.74 ± 82.32	0.149
Home moderate PA	120 ± 84.85	185 ± 103.79	156.43 ± 120.51	178.96 ± 146.65	194.17 ± 137.40	144 ± 98.79	168.42 ± 123.74	0.865
Walking	10 ± 14.14	21.67 ± 25.87	48.71 ± 51.86	21.46 ± 35.61	55.83 ± 51.60	16.50 ± 19.44	35.11 ± 44.23	0.034

Data are expressed as mean ± SD, ANOVA, *p* < 0.05 is statistically significant.

**Table 9 nutrients-14-04159-t009:** Physical activity of sample—professional (organized) physical activity.

Variable PA min/day ± DS	Underweight	Normal Weight	Overweight	Obesity Type 1	Obesity Type 2	Obesity Type 3	Total	*p* Value
Vigorous professional PA	0	0	0.86 ± 5.07	7.50 ± 36.74	0	0	2.21 ± 18.69	0.769
Moderate professional PA	0	25.00 ± 69.87	32.57 ± 109.23	0	20.00 ± 69.28	0	17.68 ± 75.22	0.636
Professional walking	0	20.83 ± 52.99	21.43 ± 83.46	2.50 ± 12.24	0	0	11.16 ± 54.65	0.677
Bicycle	0	2.50 ± 8.66	0	0	0	0	0.32 ± 3.07	0.227
Sports	0	7.50±18.64	2.14 ± 10.38	1.04 ± 3.60	0	2 ± 6.32	2.21 ± 9.55	0.449

Data are expressed as mean ± SD, ANOVA, *p* < 0.05 is statistically significant.

**Table 10 nutrients-14-04159-t010:** Multiple regression analyzes between MDA and intake of vegetables and fruit.

Model	Unstandardized Coefficients		Standardized Coefficients	t	*p*
	B	Std. Error	Beta		
(Constant)	2.093	0.107		19.545	0.000
Raw vegetables	−0.305	0.153	−0.213	−1.996	0.049
Boiled vegetables	0.030	0.171	0.019	0.177	0.860
Fruit	−0.276	0.131	−0.222	−2.100	0.039

**Table 11 nutrients-14-04159-t011:** Multiple regression analyzes between GSH oxidative marker and intake of vegetables and fruit (Dependent Variable: GSH).

Model	Unstandardized Coefficients		Standardized Coefficients	t	*p*
	B	Std. Error	Beta		
(Constant)	8.420	0.278		30.324	0.000
Raw vegetables	0.516	0.397	0.143	1.301	0.197
Boiled	0.042	0.442	0.011	0.096	0.924
Fruit	0.506	0.341	0.161	1.484	0.142

**Table 12 nutrients-14-04159-t012:** Multiple regression analyzes between GSSG oxidative marker and intake of vegetables and fruit (Dependent Variable: GSSG).

Model	Unstandardized Coefficients		Standardized Coefficients	t	*p*
	B	Std. Error	Beta		
(Constant)	1.358	0.083		16.297	0.000
Raw_vegetables	−0.166	0.119	−0.152	−1.391	0.168
Boiled_vegetables	0.050	0.133	0.041	0.376	0.708
Fruit	−0.159	0.102	−0.169	−1.557	0.123

**Table 13 nutrients-14-04159-t013:** Multiple regression analyzes between GSH/GSSG oxidative marker and intake of vegetables and fruit.

Model	Unstandardized Coefficients		Standardized Coefficients	t	Sig.
	B	Std. Error	Beta		
(Constant)	8.155	0.949		8.594	0.000
Raw_vegetables_1	1.333	1.356	0.110	0.984	0.328
Boiled_vegetables_1	0.006	1.512	0.000	0.004	0.997
Fruits_1	0.790	1.165	0.075	0.678	0.499

## Data Availability

The data presented in this study are openly available in FigShare at https://doi.org/10.6084/m9.figshare.21280257, accessed on 3 September 2022.

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
