# Peer review of "The Influence of Diet and Physical Activity on Oxidative Stress in Romanian Females with Osteoarthritis"

_nutrients, 2022, doi:10.3390/nu14194159_

Round 1

Reviewer 1 Report

In both the Introduction and the Discussion, the authors discuss the relationship between obesity and osteoarthritis, yet their findings neither prove nor disprove this relationship. Only 10 of the 98 participants had a normal weight, and the others were overweight or obese. No meaningful comparison can be made because the reference group is too small. The authors try to circumvent this limitation by subdividing the overweight/obese group into four subgroups and comparing them. But the subgroups are likewise too small for any meaningful comparison.

I suspect that the authors initially wished to examine the relationship between obesity and osteoarthritis. Unfortunately, the overall sample size is too small, and there is not enough variation in body mass index within the sample. Certainly, the authors should include the obesity data in their paper, but they should limit their discussion to their actual findings, i.e., the relationship between diet and oxidative stress markers.

Underweight women?

On the one hand, we are told that none of the women were underweight (Table 2). On the other hand, we are provided with the levels of oxidative stress markers of underweight women (Table 3) and the level of physical activity of underweight women (Table 8). How can a nonexistent group have these characteristics? Please explain. Are the underweight women from another study?

Background of participants

There is no mention of ethnicity in the demographic data. Were the participants ethnically diverse or were they all of Romanian ethnic background?

Corrections

p. 9, line 275 – replace “have showed” with “have shown”

p. 10, line 294 – delete the hyphen before “predicted pain” and replace “predicted” with “predict”

Author Response

Dear Esteemed Reviewer,

Thank you for reviewing our paper.

  • The link between obesity and OA was not the primary objective of our study. Obesity is a modifiable risk factor related to diet and lifestyle. We added this as secondary objective of the study(line 108-110).
  • Because the sample is not so large, we added this limitation in the discussion Line 293-294.
  • There was a small size of underweight women (it was a typing mistake), we corrected Table 2. This is the reason that we calculated the oxidative markers and physical activity in this class.
  • The participants had the same Romanian background.
  • We corrected line 275 with have shown, and line 294 with predict.
  • The paper was corrected again with Grammarly program for English corrections.

Kind regards,

Dr. Bogdana Nasui

From behalf of the authors

Reviewer 2 Report

The manuscript was prepared very well. However, there are some concerns, in part important, so the review articles need revision, see below.

General comments

·       The numbering of the references in the text are not correct (1,2,3); the correct form is (1-3)

·       Include previous history of similar investigations and justify the need for this investigation

Materials and Methods / Results

·       This is the strong part of the study; I congratulate the authors.

Discussion

·       It should include some hypothesis of the possible mechanisms described from a physiological perspective or include some mechanism of action described.

·       It should include some comparative discussion with other studies related to its purpose, explaining the differences.

·       Include a section on strengths

·       What does this study contribute? Clarify.

·       Any possible application of the results described?

·       Include a section on strengths and limitations.

Author Response

Dear Esteemed Reviewer,

Thank you for reviewing our paper.

  • In the introduction, we justify the need of investigation (line 99)
  • In the discussion section, we included possible mechanisms (line, 327-333, line 368-395)
  • Comparison with other human studies from literature was difficult to make because the data from literature are missing, regarding studies from human blood samples. We included this in our discussion.
  • We modified the references according to the journal requirements.
  • We included the strength of our study and possible application (line 400-407) .
  • Limitations of the study are included (line 417-424).
  • Our study is one of the few studies that open the perspectives and correlation between OA, oxidative and nutrition and lifestyle., as we mentioned in the paper (line 402).
  • English language correction was made using the Grammarly program.

Kind regards,

Dr. Bogdana Nasui
